# Effects of Compression Stockings on Body Balance in Hemiplegic Patients with Subacute Stroke

**DOI:** 10.3390/ijerph192316212

**Published:** 2022-12-04

**Authors:** Eo Jin Park

**Affiliations:** Department of Rehabilitation Medicine, Kyung Hee University Hospital at Gangdong, Seoul 05278, Republic of Korea; cp1024@naver.com; Tel.: +82-2-440-7246

**Keywords:** hemiplegia, stroke, body balance, compression stocking

## Abstract

(1) Background: Stroke patients with hemiplegia have an increased risk of developing deep vein thrombosis (DVT). DVT increases the risk of life-threatening pulmonary embolism and is associated with poor prognosis. The early wearing of compression stockings can help prevent DVT. This study aimed to assess the impact of compression stockings on body balance in stroke patients with unilateral lower extremity muscle weakness; (2) Methods: Hemiplegic stroke patients in the subacute phase who were able to walk with assistance were recruited. The patients were divided into two groups: one group received rehabilitation treatment with compression stockings, and the other received treatment without compression stockings. The rehabilitation treatment involved hospitalization for 4 weeks, the Trunk Control Test (TCT), the Trunk Impairment Scale (TIS), and the Berg Balance Scale (BBS). The patients were evaluated before and 4 weeks after the start of treatment. The differences in BBS, TCT, and TIS before and after treatment between the two groups were compared; (3) Results: Altogether, 236 hemiplegic stroke patients were recruited. There was an improvement in body balance after treatment in both groups, and BBS, TCT, and TIS scores significantly increased in the group that received rehabilitation treatment with compression stockings; (4) Conclusions: In patients with hemiplegic stroke in the subacute period, rehabilitation while wearing compression stockings appears to improve body balance.

## 1. Introduction

Worldwide, more than 12.2 million new stroke cases occur each year, and it is reported that 25% of the population over the age of 25 experiences a stroke [1]. In Asia, the incidence of stroke is reported to be 67–422 per 100,000 person-years [2]. After a stroke, individuals have varying degrees of loss of sensory, motor, and upper-brain cognitive abilities, which impairs balance [3]. It has been shown that hemiplegic stroke patients have increased sway in their posture, asymmetrical distribution of weight, and poor weight shifting [4]. According to several studies, fall prevalence is reported to be approximately 14–73% worldwide [5,6,7] and approximately 24–37% in Asia [8,9] during the first 6 months after the stroke episode. Furthermore, poor balance increases the risk of falls [10]; falls can cause cerebral hemorrhage, multiple bone fractures, and femoral neck fractures, which can worsen the patient’s clinical outcome and lead to death in severe cases. In particular, since motor weakness is severe and the risk of falls is high in the early stages after stroke onset, effective management is required to improve balance in the subacute period [11]. Treatment options for improving balance are important to reduce mortality after stroke and improve the patient’s functional outcome.

A higher incidence of thromboembolism and unilateral lower extremity edema has been linked to hemiplegia after stroke [12]. In acute stroke patients, compression stockings are regularly prescribed for the prevention of deep vein thrombosis [13]. Other benefits of compression stockings include an increased anaerobic threshold when walking, increased power output after fatigue, and enhanced post-exercise recuperation [14,15]. One of the most significant physiological effects of compression stockings is a change in sensory feedback. Several studies have addressed the possible consequences of altered sensory input from compression stockings on motor functions. The use of compression stockings increases power production during repeated activities [16]. In addition, the effects of reducing muscle oscillation and improving joint recognition have been reported in several studies [14,16]. Improvements in visuomotor tracking tasks have also been reported when compression stockings are worn [17]. The suggested mechanism for these effects is the modification of group Ia presynaptic inhibition in the spinal cord [18]. It is assumed that this interaction occurs between passively administered sensory enhancements and conventionally measured spinal cord reflex circuits [18]. Sensory input from cutaneous receptors is essential for modifying muscle activity to adjust to environmental changes and avoid falling during locomotor activities [19,20]. Numerous studies on cutaneous reflexes in the leg have shown task- and phase-dependent reflex regulation [21]. The general and specialized processes involved in controlling the stance and swing phases of gait are influenced by sensory feedback from specific skin regions on the foot [17]. Another study found that using compression stockings significantly reduced postural sway and improved balancing time [22].

The purpose of this study was to determine whether wearing compression stockings on the hemiplegic side could help improve body balance in stroke patients.

## 2. Materials and Methods

### 2.1. Subjects

Between January 2012 and May 2022, stroke patients who were admitted to the Department of Rehabilitation Medicine at Kyung Hee University Hospital in Gangdong within two weeks of stroke onset were retrospectively recruited. Patients with hemiplegic side lower extremity strength of Medical Research Council (MRC) grade 3 or lower participated in this study [23]. Patients who were capable of standing and walking with assistance were asked to perform a balance test. To ensure the necessary level of cognitive function to follow instructions, patients with a Mini-Mental State Exam (MMSE) score of 20 or higher were included. Patients with brain stem lesions that could cause dizziness and equilibrium problems that could affect balance and patients with a history of previous stroke, autonomic neuropathy, benign paroxysmal positional vertigo, and orthostatic hypotension were excluded. The patients were divided into two groups. Patients with skin lesions that prevented wearing of compression stockings, such as pressure ulcers or contact dermatitis, or patients who refused to wear compression stockings due to discomfort were assigned to Group B (Figure 1). Group A wore compression stockings (SIGVARIS, St. Gallen, Switzerland) with an interface pressure of 23–32 mmHg, made with nylon blended fabric that reached above the knee on the hemiplegic side, for the whole day; they also received conventional rehabilitation. Group B received conventional rehabilitation without compression stockings. The study protocol was approved by the Institutional Review Board (IRB) of Kyung Hee University Hospital in Gangdong (IRB approval number: 2022-10-038).

### 2.2. Procedure

The conventional rehabilitation program was conducted for 90 min a day, 5 times a week, for 4 weeks. The conventional rehabilitation program consisted of physical therapy, occupational therapy, and neurodevelopmental treatment [24]. The treatment program included trunk, balance, sitting, standing, and walking exercises [25]. In addition, abdominal muscles, core muscles, lower extremity strengthening, proprioceptive training in balance performance, trunk control, lower extremity weight shifting, sitting and standing balance training, and squatting exercises were instituted [25,26]. During rehabilitation, the patients continued to wear compression stockings. The Berg Balance Scale (BBS), Trunk Control Test (TCT), and Trunk Impairment Scale (TIS) were evaluated by a physical therapist and a physical medicine and rehabilitation specialist before and after the start of treatment to measure balance.

### 2.3. The Berg Balance Scale

The BBS is a common clinical test used to evaluate a person’s dynamic and static balance functions. To assess the balance status in various postures, the BBS comprises a series of 14 functional balancing tasks, such as sitting to standing, transferring weight and reaching, standing on one leg, turning in place, and sustaining a tandem stance [27]. Each task was scored on a scale of 0–4. A score of 0 indicated an inability to accomplish the task, while a score of 4 indicated the capacity to carry it out in accordance with a predetermined standard. The maximum score is 56; the greater the score, the better the balancing function [28]. The sensitivity of the BBS for predicting fall risk was reported to be 72%, with a specificity of 73% [29,30]. The intraclass correlation coefficient for test–retest reliability was 0.99, and Cronbach’s alpha was reported with a reliability of 0.97 [29]. An initial evaluation was conducted before starting rehabilitation treatment, and re-evaluation was performed after 4 weeks of rehabilitation treatment.

### 2.4. The Trunk Control Test Score

The TCT performs an assessment of the following movements: rolling from a supine position to a weak side and a strong side, standing from a supine position, and sitting with the feet off the floor for 30 min in a balanced position on the edge of the bed [31]. Zero points are given if the patient is unable to move without assistance, 12 points if the movement can be performed but in an abnormal way, and 25 points if the movement can be completed normally. TCT scores are summed from a minimum of 0 to a maximum of 100, with higher scores indicating a better balance function [31]. The validity of the TCT was reported as having 98% sensitivity and 92.2% specificity [32]. The intraclass correlation coefficient for test–retest reliability was 0.979, and Cronbach’s alpha was reported with a reliability of 0.749 [33]. An initial evaluation was conducted before starting rehabilitation treatment, and re-evaluation was performed after 4 weeks of rehabilitation treatment.

### 2.5. The Trunk Impairment Scale

The TIS involves the evaluation of dynamic sitting balance, static sitting balance, and coordination [34]. The dynamic sitting balance score was evaluated with a maximum of 10 points, the static sitting balance score with a maximum of 7 points, and the coordination score was evaluated with a maximum of 6 points. The scores range from a minimum of 0 points to a maximum of 23 points. The higher the score, the better the balance function [34]. The validity of the TIS was reported to have 78% sensitivity and 85% specificity [35,36]. The intraclass correlation coefficient for test–retest reliability was 0.964, and Cronbach’s alpha was reported with a reliability of 0.982 [36]. An initial evaluation was conducted before starting rehabilitation treatment, and re-evaluation was performed after 4 weeks of rehabilitation treatment.

### 2.6. Modified Barthel Index

The Modified Barthel Index (MBI) is a scale that evaluates a patient’s ability to perform activities of daily living. It consists of 10 items: eating, dressing, grooming, toileting, bladder control, bowel control, stair climbing, ambulation, chair transfer, and bathing [37]. A total of 100 points is the highest score, and higher scores mean that it is possible for the patient to perform daily tasks more independently. The assessments were performed before the patients underwent the rehabilitation treatment.

### 2.7. Statistical Analysis

All of the statistical studies used SPSS version 25.0 (IBM Corp., Armonk, NY, USA), and the Kolmogorov–Smirnov test was performed to assess the normality of the data. Levene’s test was used to determine the homogeneity of variances across groups. The continuous variables were analyzed using the independent *t*-test, and the categorical variables were analyzed using the chi-square test. The continuous variables within a group were compared using paired *t*-tests. The significance level for all statistical tests was set at *p* < 0.05.

## 3. Results

### 3.1. Subjects

Altogether, 236 patients were included in this study, comprising 115 men and 121 women, with a mean age of 69.55 ± 12.92 years. Among them, 126 patients had ischemic stroke, and 110 patients had hemorrhagic stroke. The average Modified Barthel Index (MBI) of the patients was 55.31 ± 8.7, and the average score of the MMSE was 22.99 ± 3.22. The average BBS was 22.00 ± 4.57, the average TCT score was 40.13 ± 6.11, and the average TIS was 10.56 ± 1.65, indicating a decrease in balance (Table 1). There was no statistically significant difference between the two groups in terms of age, sex, type of stroke, MBI, MMSE, BBS, TCT, or TIS, measured before the start of treatment (Table 2).

### 3.2. Comparison of Balance Function before and after Treatment

A comparison of the balance function before and after 4 weeks of treatment within the same group showed that in group A, BBS increased from 21.89 ± 4.44 to 31.97 ± 5.86, TCT increased from 40.05 ± 6.35 to 55.37 ± 9.90, and TIS increased from 10.54 ± 1.72 to 14.86 ± 3.20 and all showed statistically significant results. In group B, BBS increased from 22.13 ± 4.74 to 27.30 ± 5.92, TCT increased from 40.23 ± 5.82 to 48.06 ± 8.83, and TIS increased from 10.57 ± 1.57 to 12.98 ± 1.89, all of which showed statistically significant results (Table 3).

A comparison of the changes in the balance function score before and after 4 weeks of treatment between the two groups showed that group A, which underwent rehabilitation treatment with compression stockings, had a significantly higher balance function than that of group B, which underwent treatment without compression stockings (Table 4).

## 4. Discussion

The purpose of this study was to investigate whether wearing compression stockings during rehabilitation therapy improves balance in patients with subacute hemiplegic stroke. Rehabilitation treatment was performed for 4 weeks in hemiplegic stroke patients with impaired balance function, and the results showed that balance function improved in both the group wearing compression stockings and the group not wearing them. A greater improvement was observed in the group that wore compression stockings. These results suggest that rehabilitation treatment while wearing compression stockings during the subacute period in patients with hemiplegic stroke may help improve body balance.

In the body balance system, motor and sensory functions, such as proprioception, are both involved in the maintenance of balance [38]. Several studies, initially looking at improving motor function, have reported the effectiveness of compression stockings during exercise. An earlier study showed that compression stockings helped with submaximal intensity in physiological domains such as improving oxygen use, increasing blood flow, and reducing muscle oscillation [14]. Compression stockings have also been shown to successfully prevent delayed-onset muscle soreness after sessions of maximal-intensity exercise [39]. In addition, the use of compression stockings appears to influence the post-exercise levels of lactate clearance in the blood [14]. The effect of compression stockings on improving muscle blood flow and muscle fatigue recovery suggests that motor-functional outcomes may be improved during exercise therapy after stroke [14,15].

In order to maintain equilibrium, the central nervous system combines vestibular, visual, and proprioceptive information to create motor orders that coordinate muscle activation patterns [40]. Proprioception is described as the capacity to integrate sensory inputs from numerous mechanoreceptors to identify body position and movement in space, and it plays a critical role in balance regulation [41,42]. Decreased sensitivity of peripheral mechanoreceptors in the joints, skin, and muscles is mainly responsible for the decline in proprioception acuity [43]. A few prior studies have shown that compression stockings may increase joint proprioception and visuomotor movement precision [44]. Using an ecological dynamics paradigm, the process underlying the augmentation of proprioceptive information while wearing compression stockings has been described. It is reported that wearing compressive materials that contort, compress, and deform receptors on the joint soft tissue and skin surface can increase sensorimotor system noise [45]. These materials may counterintuitively aid performers in enhancing their sense of somatosensory feedback by boosting information from background noise [46]. Cutaneous sensors in the lower extremities provide information for joint location and play an essential role in supplying kinesthetic data during joint matching exercises [47]. Compression stockings were shown to improve joint accuracy and sensitivity, as well as the direction sense of the extremities during movement, by filtering out incorrect mechanoreceptor input and improving proprioceptive sensory information [48,49]. These effects on proprioception suggest that compression stockings may help improve balance after stroke [50,51]. Patients with impaired balance are at higher risk for falls. The risk of falls is particularly high in the subacute period, and active intervention is required to improve balance during this period [52]. Therefore, the results of this study suggest that not just conventional rehabilitation but rehabilitation treatment while wearing compression stockings on the hemiplegic side can reduce the risk of falls and help improve balance. What patients can do at home to improve their functional outcomes after stroke is rather limited. However, because it is not too difficult to wear compression stockings, they can be a viable treatment option for patients at home. Based on the results of this study, when living at home after a stroke, wearing compression stockings and undergoing rehabilitation treatment can improve body balance.

This study has a few limitations. First, this was a nonrandomized, retrospective study. Many patients complain of discomfort while wearing compression stockings, and it is difficult to continue wearing them. Therefore, it is possible that the group that wore compression stockings for 4 weeks had a greater balance function improvement than the group that did not wear them due to higher compliance and participation in the rehabilitation treatment. Second, only BBS, TIS, and TCT were used as indicators to evaluate body balance. Although these three indices are useful and widely used tools to assess body balance, other balance scales, such as the timed up and go [53], dynamic gait index [53], and Tinetti test [54], have not been evaluated. Finally, by comparing the results before and after 4 weeks of rehabilitation treatment, a short-term effect was observed in the subacute period, but the long-term effect was not confirmed. A prospective, randomized, controlled study is required in the future. 

## 5. Conclusions

Compression stockings may be an option to improve the body balance of patients with hemiplegic stroke in the subacute period. Our results may also provide objective evidence for the effectiveness of the use of compression stockings in improving body balance in patients with subacute hemiplegic stroke.

## Figures and Tables

**Figure 1 ijerph-19-16212-f001:**
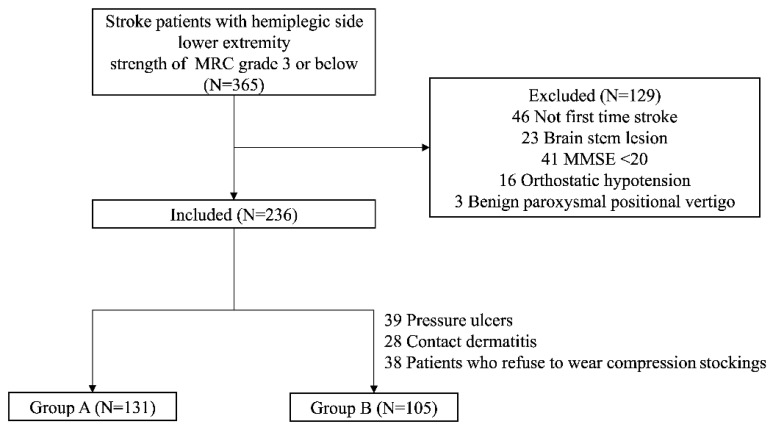
Flow chart of study participant enrollment.

**Table 1 ijerph-19-16212-t001:** Clinical characteristics of the patients.

Characteristic	Value
Age (years)	69.55 ± 12.92
Sex	
Male	115 (48.70)
Female	121 (51.30)
Type of stroke	
Ischemic	126 (53.40)
Hemorrhagic	110 (46.60)
MBI	55.31 ± 8.79
MMSE	22.99 ± 3.22
BBS	22.00 ± 4.57
TCT	40.13 ± 6.11
TIS	10.56 ± 1.65

Values are presented as mean ± standard deviation or number (%). MBI, Modified Barthel Index; MMSE, Mini-Mental State Examination; BBS, Berg Balance Scale; TCT, Trunk Control Test; TIS, Trunk Impairment Scale.

**Table 2 ijerph-19-16212-t002:** Baseline characteristics of the two groups.

Variables	Group A (*N* = 131)	Group B (*N* = 105)	*p*-Value
Age (years)	70.79 ± 12.41	68.75 ± 13.06	0.24
Sex			0.13
Male	58 (44.30)	57 (54.30)	
Female	73 (55.70)	48 (45.70)	
Type of stroke			0.99
Ischemic	70 (53.40)	56 (53.30)	
Hemorrhagic	61 (46.60)	49 (47.70)	
MBI	55.87 ± 8.82	54.60 ± 8.74	0.27
MMSE	22.89 ± 3.15	23.11 ± 3.32	0.32
BBS	21.89 ± 4.44	22.13 ± 4.74	0.68
TCT	40.05 ± 6.35	40.23 ± 5.82	0.82
TIS	10.54 ± 1.72	10.57 ± 1.57	0.89

Values are presented as mean ± standard deviation or number (%). MBI, Modified Barthel Index; MMSE, Mini-Mental State Examination; BBS, Berg Balance Scale; TCT, Trunk Control Test; TIS, Trunk Impairment Scale.

**Table 3 ijerph-19-16212-t003:** Change of balance function by treatment within the group.

Variables	Group A (*N* = 131)			Group B (*N* = 105)		
	Pre	Post	*p*-Value	Pre	Post	*p*-Value
BBS	21.89 ± 4.44	31.97 ± 5.86	0.031 *	22.13 ± 4.74	27.30 ± 5.92	0.027 *
TCT	40.05 ± 6.35	55.37 ± 9.90	<0.001 **	40.23 ± 5.82	48.06 ± 8.83	<0.001 **
TIS	10.54 ± 1.72	14.86 ± 3.20	0.015 *	10.57 ± 1.57	12.98 ± 1.89	0.031 *

Values are presented as mean ± standard deviation. BBS, Berg Balance Scale; TCT, Trunk Control Test; TIS, Trunk Impairment Scale. * *p* < 0.05, ** *p* < 0.001.

**Table 4 ijerph-19-16212-t004:** Change of balance function before and after treatment between the two groups.

Variables	Group A (*N* = 131)	Group B (*N* = 105)	*p*-Value
ΔBBS	10.08 ± 6.94	5.17 ± 7.32	0.015 *
ΔTCT	15.31 ± 11.58	7.82 ± 10.69	<0.001 **
ΔTIS	4.32 ± 3.47	2.40 ± 2.56	0.032 *

Values are presented as mean ± standard deviation. ΔBBS, The difference between before and after treatment on the Berg Balance Scale; ΔTCT, The difference between before and after treatment on the trunk control test score; ΔTIS, The difference between before and after treatment on the trunk impairment scale score. * *p* < 0.05, ** *p* < 0.001.

## Data Availability

The datasets generated and/or analyzed during the current study are available from the corresponding author upon reasonable request.

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
