# Peer review of "Effects of Compression Stockings on Body Balance in Hemiplegic Patients with Subacute Stroke"

_ijerph, 2022, doi:10.3390/ijerph192316212_

Round 1

Reviewer 1 Report

Thank you for allowing me to review this manuscript. This study has good potential for clinical and scientific applications, but the methodology has many limitations that need revision.

- In the introduction, I recommend: 1. To show the frequency of falls in this population in the world and Asia countries; 2. use recent references about stroke (it would be helpful to include the actual number of strokes worldwide and/or number of strokes in Asia)

 - In the methods: overall, the organization and content did not flow well.

a) The recruitment period was ten years, to explain why.

b) To show the sample size calculation or present a flowchart with all information on the number of inclusion and exclusion throughout the study period.

c) How did the patients allocate in each group? Randomization?

d) Please provide a script or details on the method of collecting (team of investigators?

Who did apply the scales? Who did evaluate before and after the intervention?

e) Did all the data fit a normal distribution? More information about the analysis concerning the data and variables would be helpful.

f) in the results, it was presented the Barthel Index values: explain this information in the methods. When did the scale use?

g) Scales: please, show all scales and specifically state when the scales were administered to the patients. Also, show the values of validity and reliability of each scale. 

h) Were the Equator guidelines followed?

 - Resuts: ok

- Discussion: It would be helpful to have more explanation about the relations and measures that could indicate the relevance of these findings for rehabilitation professionals.

- Complete the Limitations.

- References: Most references are older than 5 years. The selection of references compromises the discussion because out of 36 references, and only 5 were published in the last 5 years. Stroke care has changed over this period of the study.

Author Response

Thank you again for your valuable reviews and suggestions.

Reviewer 2 Report

The purpose of the study was to determine whether wearing of compression stockings on the hemiplegic side can help improve body balance in pre-acute stroke patients.   Quality of life, rehabilitation and prevention of complications are important parts of care for patients  therefore the topic of the study we can consider as an important. The author provided comprehensive literature review for justification of the topic, clear sampling procedure. In subsection 2.2 (lines 75-84) I did find information about class of compression for stockings, material (fabric), how (both sides or only hemiplegic) and how long patients wore stockings (whole day, only during rehabilitation procedure or other regimen of wearing). It would be useful, to my mind, for other researchers for replication and further research. And one more doubtful moment indication of the brand of stockings from the ethical point of view but probably the author could justify it.

The results of the study provided in the manuscript very clear and evidence-based, discussion part is useful and interesting for clinicians as well as for public health specialists. Conclusion is clear, references are in accordance to the topic of the study.

Author Response

(The authors gave the same response as above.)

Reviewer 3 Report

Introduction:

·       Improved the introduction. The main reason to perform the study is not clear.

Materials and methods:

·       The time between the stroke and the recruitment must be defined. The concept of “subacute stroke” is not clear as inclusion criteria.

·       How many times the subjects of group A used the compression stockings? It is a possible confounding variable that must consider in the analysis.

·       Clarify the sampling assignation. It was a random assignment?

·       Describe the rehabilitation program.

·       How is the statistical inference guaranteed considering that the sample size was not calculated?

Results:

·       Only comparation were made with the variables included? Other variables have to been considering in the comparation between groups (table 2).

·       T-test is the correct statistical test for the distribution of the data?

Discussion:

·       The discussion must be improved. The paragraphs seem disjointed and some of the information was commented on in the introduction.

Conclusion:

·       The author overestimates the results. The study does not allow infer the effectiveness of the compressed stockings improvement the balance.

Author Response

(The authors gave the same response as above.)

Round 2

Reviewer 1 Report

Thank you for allowing me to review this manuscript for the second time. 

All sugestion were accepted by the authors. I have only one more consideration about the themes of the Barthel Index. 

Sinecerelly

Author Response

(The authors gave the same response as above.)
